# The Value and Potential Integration of Pharmacy Technician National Certification into Processes That Help Assure a Competent Workforce

**DOI:** 10.3390/pharmacy7040147

**Published:** 2019-11-05

**Authors:** Shane P. Desselle, Kenneth C. Hohmeier, Kimberly C. McKeirnan

**Affiliations:** 1College of Pharmacy, Touro University California, Vallejo, CA 94592, USA; 2College of Pharmacy, University of Tennessee Health Sciences Center College of Pharmacy, Nashville, TN 37211, USA; khohmeie@uthsc.edu; 3College of Pharmacy, Washington State University, Pullman, WA 99202, USA; kimberly.mckeirnan@wsu.edu

**Keywords:** technician, pharmacy, certification, education, preparedness

## Abstract

The purposes of this study were: (1) to determine pharmacists’ perceptions of the impact of certification on competence in specific job skills, its impact in combination with job experience, and its impact in combination with other types of vocational education/training; (2) to identify elements that could potentially enhance the value, or impact of national certification; and (3) to determine how pharmacists view certification in light of various personnel management and organizational behavior phenomena. A self-administered survey was constructed and delivered in spring of 2019 to a random sample of four U.S. states chosen for their geographic diversity and relatively high proportions of both certified and non-certified pharmacy technicians. Following multiple reminders, a response rate of 19.3% was obtained. The 326 responding pharmacists saw certification being less impactful alone than when combined with other types of education/training and previous job experiences. They saw the need for more skills-related and “soft skills” content on the certification examination and agreed that certification is a factor in hiring decisions and that it should be required for designation for advanced practice status. Taken together, respondents saw the need for pharmacy leaders to integrate certification with other aspects of preparation to make for a more competent and professional workforce support team.

## 1. Introduction

Pharmacy technicians and other workforce support personnel are recognized as essential in the evolution of pharmacy practice to a more patient-centric focus and public health orientation [1]. The previous few years are witness to considerable research into pharmacy technician roles, moving beyond descriptions of technician practice in one particular facility into broader examinations of roles that can be consistently delegated across organizations and even practice settings [2,3,4,5]. The growth in literature has helped spur recent systematic reviews of pharmacy technician practice. In one such review, Mattingly and Mattingly noted that approximately half of studies on pharmacy technician practice had been published in the previous decade and that on-the-job training was allowing them to assume more administratively based positions [6]. They found the benefits to technicians for these shifts in practice to be more indirect and/or intrinsic, thus associated with very little raises in pay. Another review centered more around uptake of specific roles associated with pharmacist provision of medication therapy management (MTM) services [7]. That review included 44 manuscripts describing pharmacy technician involvement with medication reconciliation (70% of papers reviewed), documentation (41%), medication therapy review (30%), medication record development (5%), physical assessment (5%), and patient follow-up (2%). The authors concluded that standardized training for pharmacy technicians that delineates administrative support from pharmacists’ role of clinical decision making could help pharmacists achieve greater efficiency in MTM delivery.

These reports in the literature evince significant strides in technician practice but that which still stands much room for further growth and improvement. There have been a number of pleas for greater technician involvement in various roles such as telephonic prescription transfer [8], immunizations [9], and quality assurance [10]. However, technician work is already reported to be stressful [11] and for relatively little pay that does not improve with greater regulatory requirements for registration and/or licensure in the United States (U.S.) [12] Aside from assistance with medication reconciliation, much of the attention and growth of technician practice responsibilities have been under the auspices of re-engineering models such as “tech-check-tech” or “technician product verification” where technicians are delegated more tasks in the dispensing process and in some cases afforded a considerable amount of authority in supervising one another’s work up to the point where the prepared medication order is provided to the patient [13]. Even while the presence of such “checking technicians” has been demonstrated to be safe, there is some reluctance in advancing technician roles much further [14].

At least a significant if not primary reason for this reluctance is the lack of standardization, even agreement, on the education, training, and professional development necessary for entry into practice and continued employment and advancement [15]. Leaders in pharmacy have long called for national (U.S.) standards for technician education, training, state licensure, as well as for properly defined “entry-level” versus “advanced” technicians [16,17]. The momentum for such clarity in standards has gained even more traction following a stakeholder consensus meeting of pharmacy leaders from various settings and from agencies with regulatory authority [18]. However, the current picture still sees wide variation in technician training and credentialing requirements from state to state. Entry-level practice requirements for technicians throughout the nation indicate that just over half of U.S. states require no education/training or certification of any type; five require certification, only; four require education/training only but not certification; seven require some sort of education/training and certification; another seven require either education/training or certification; and seven have no requirements for education, registration or licensure [19].

Given the lack regulatory authority by one national body, achieving further clarity on entry-level and advanced practice has remained elusive [20]. Moreover, various stakeholders, including some large employers might favor the status quo [21]. In addition to concern about rising labor costs in the face of tight profit margins, some employers might have preference for on-the-job training that fits their organization’s specific requirements for the jobs they have designed.

Despite these factors, many stakeholders have embraced national certification as administered by either the Pharmacy Technician Certification Board (PTCB, administering the PTCE^®^ examination) or through the National Healthcareer Association (NHA, administering the ExCPT examination). Several U.S. states now require certification for pharmacy technicians for registration and/or licensure. Other states are considering adopting the requirement for entry or for the designation of so-called “advanced” or similar such designations [22]. The state of Washington requires national certification in addition to experiential work in several, mandatory areas of knowledge [23]. Additionally, some employers have begun to mandate certification to coincide with in-house training, even requiring that PharmD students working as interns acquire certification, as well.

Both national certification procedures involve a self-study process culminating in an examination with components in the names and indications of common drugs, basic pharmacology, federal jurisprudence, dispensing processes, compounding, sterile intravenous admixture, medication safety/quality assurance, and issues surrounding controlled substances [24,25]. Impending changes examination suggest a more parsimonious set of domains, for example removal of sterile intravenous admixture from the PTCE [24].

The principle aims of certification have thusly been on imparting essential knowledge in carrying out the duties of a pharmacy technician. There is no experiential component or skills-based assessment. However, evidence suggests that engagement in the certification process imbues a greater sense of professional identity and thus might spur greater professionalism and greater commitment to a pharmacy career [26].

While various stakeholders debate requirements for education and training of pharmacy stakeholders, it is important to discern the value of national certification. Previous studies on the value of certification were conducted approximately a decade ago [27,28]. These studies found modest contribution of certification toward various skills and attitudes. Much has changed since the publication of those studies, and that research was conducted in the absence of context, or consideration of other types of education and training. That is, those studies did not determine the extent to which certification might assist or be leveraged during other components or possible education training modalities, such as vocational education and on-the job training.

To that end, the overall aims of this study were to ascribe value to the certification process, and specifically: (1) to determine pharmacists’ perceptions of the impact of certification on competence in specific job skills, its impact in combination with job experience, and its impact in combination with other types of vocational education/training; (2) to identify elements that could potentially enhance the value, or impact of national certification; and (3) to determine how pharmacists view certification in light of various personnel management and organizational behavior phenomena.

## 2. Materials and Methods

### 2.1. Survey Design

The study methods were deemed exempt from full evaluation and approved for conduct by the principal investigator’s Institutional Review Board (IRB).

The study employed a cross-sectional design with use of a survey targeted to a sample of pharmacists from four U.S. states. The survey was comprised of several components. In the first component, pharmacist respondents were asked to identify the impact of certification, alone, on a technician’s competence in performance of 21 different job functions and responsibilities in accordance with previous job analyses [29] but in this case job behaviors and roles that were not unique to a particular setting and that comported with components of professionalism and an organization behavior framework proposed by Roberts et al. [30] in pharmacy settings. This organizational behavior framework provides a useful perspective for recognizing the contributions of constituents within an organization, and in turn how their behaviors may affect each other and the organization as a whole. The items used for scaling, then, were items such as prescription/medication order entry, medication preparation, compounding, billing, supervision of other technicians, problem-solving, leadership, time management, basic pharmacology, math skills, and ability to adapt to organizational change. These items were evaluated on a five-point Likert-type scale ranging from 1 = Not all, to 5 = Very much. Participants then evaluated the impact of certification using the same scale and same items but this time in combination with experience on the job, and then also evaluated those same items using the same scale but for certification’s impact in combination with other types training/education, such as vocational and on-the-job training.

The next component of the survey contained eight items representing possible actions taken to improve the utility or increase the value of certification. The items were scaled on importance on three points from “Not Important” to “Somewhat important” and “Very important”. In this case, the investigators did not believe in the necessity of additional scale intervals and preferred to keep the scale simple and balanced between the intervals (hence, three points only). The items included various components such as more content in specific areas, more support from employers, more specialty certifications, better integration of the examination with vocational education, and more stringent criteria to be eligible to sit for the certification examination. These items were taken from the literature expressing potential improvements made to educational and training mandates for technicians, dating back to older calls and to the more recent aforementioned consensus gathering [16,17,18].

The third component of the survey to further assist in ascribing value to national certification included 13 items evaluated on five-point Likert-type scales of agreement asking respondents their opinions on various items such as the extent to which certification assists newer versus more experienced technicians, the extent that certification is associated with more greater employer commitment, whether certification should be required for advanced status, whether it helps prepare technicians for emerging roles (practice and organizational change), and whether certification is a determining factor in hiring decisions. These items were based upon findings from a previous nationwide survey of certified pharmacy technicians that examined descriptively pharmacy technicians’ commitment levels and from previous studies of pharmacists who initially ascribed value to national certification based upon technicians’ preparedness for entry-level practice at that time [27,28,29].

The fourth component of the survey asked responding pharmacists to rank eight potential characteristics or experiences of technicians as to their importance in that hiring decision. The items consisted of whether they were certified, previous work experience, anticipated job abilities, communication skills, emotional intelligence, and ability to adapt to practice change. As there were no previous studies on desired skills of technicians, these were adapted from a recent study of desire skills for pharmacists [31]. The fifth and final component of the survey solicited certain respondent demographic and practice setting characteristics.

### 2.2. Design and Sampling

The survey was constructed and disseminated using Qualtrics XM [32] and delivered via email to potential respondents. A list of pharmacists’ email addresses was acquired from IQVIA, a company that among other things maintains a list of pharmacists who have agreed to be maintained on a list of theirs to potentially be contacted for research and other purposes. There was no formal sample size determination, as there were many potential variables upon which to base power. It was hoped to acquire at least 200 or more respondents, and the study’s budget provided the purchase of 1800 emails from IQVIA.

The sampling frame was derived from four states: California, Florida, Tennessee, and Ohio. These states were selected in consultation with the study sponsor (PTCB) in identifying states that were geographically diverse with varied scopes of practice and licensure for pharmacy technicians. Concurrently, and even more importantly, these four states were without requirements for certification but still had relatively large proportions of technicians who were certified, thus yielding a greater likelihood that pharmacists would have had an opportunity to work with both certified and non-certified technicians, and particularly technicians who had been newly certified as to provide context and potential comparison for their answers to the survey. The states were sampled in relative proportion to the size of their technician population, but with some under sampling from California and some oversampling from Tennessee and Ohio to better assure a reasonable number of respondents from these states. The total number of participants contacted for participation was: 600 from California, 500 from Florida, 360 from Ohio, and 340 from Tennessee.

The procedures employed techniques recommended by Dillman et al. [33] to optimize survey response. An initial email notification of the upcoming survey was sent in early April 2019. Approximately one week later, an email with basic purpose and IRB approval (cover letter) was emailed with a link to the survey. Two reminders were sent via email to the entire sample (not knowing who had already responded) approximately one week apart, with the survey having been closed on 23 May 2019.

### 2.3. Analysis

Descriptive statistics were tabulated and reported here. There were no measures or other bases around which to frame any attempt to discern construct validity, as there were no attempts to create a summated scale score. However, internal consistency reliability was discerned among the various survey components by calculating Cronbach’s alpha scores.

## 3. Results

### 3.1. Response Rate and Respondent Characteristics

Of the 1800 survey links disseminated, 110 were returned with undeliverable email addresses. There were valid responses from 326 respondents, resulting in a response rate of 19.3%. Response rates by U.S. state (assuming an equal proportion of undeliverable surveys for each) ranged from a low of 14.6% for Florida to a high of 21.3% for Tennessee. Just over 2/3 of respondents were White/Caucasian (see Table 1), and just over 1/8 indicated a preference not to answer. Nearly 3/4 of respondents worked the equivalent of full-time hours (i.e., greater than 39 h). Over half of the responding pharmacists came from the community pharmacy setting, with an approximately equal share among those from independent and chain settings, respectively. Just under 1/4 came from hospital/health-system settings. There were a considerable number (nearly 7%) who came from a compounding or other specialty practice, and several apiece representing various other practice settings. Staff pharmacists represented nearly 1/3 of respondents, while several respondents were in some sort of administrative or ownership position. Clinical pharmacists could have come from any of various settings, but many of the pharmacy managers likely came from community settings with administrative responsibilities in addition to staffing those pharmacies.

In comparison to the general population of pharmacists in the U.S., the Bureau of Labor Statistics only categorizes pharmacists into much broader work settings, with 26% working in hospital (compared with the current study’s 23%) and 57% working in retail (compared with the current study’s 54%), which is commensurate given that some of the current study respondents likely work in a hospital or retail setting but have a job title/responsibilities that might be more clinical or administrative [34]. A 2014 study of a national random sample of pharmacists also responding to a survey showed responses in their sample from 56% who were female and 44% male, compared with the current study of just over 52% female and just under 48% male [35].

### 3.2. Survey Results

Cronbach’s alpha calculations for all subsets of items ranged from a low of 0.83 to a high of 0.97. Respondents’ perceptions of the impact of certification on technician competence alone, or in combination with other types of education/training and with previous work experiences are described in Table 2. Certification alone was not deemed to have a very substantial impact on many of the general skills under question that generally transcend most, if not all practice settings. Some of the skills/items where certification alone was rated as having the least impact were billing/administrative functions, time management skills, leadership, and problem-solving. Although still under the median scale value (“3”), the areas in which certification alone was deemed to have greater impact were basic pharmacology/drug knowledge, math computation, medication order/prescription entry, and non-sterile compounding. Respondents were more positive about the impact of certification in combination with other types of education/training, with nearly all response means calculated to be at or above the median scale value (except for time management). In addition to time management, those items/areas where the impact of certification in combination with additional education/training was rated lowest included interpersonal communication, ethical decision making and managing organizational change. Those areas rated highest included mathematical computation and medication/prescription order entry, but also sterile compounding. Some of the larger incremental evaluations from combination of certification with education/training versus certification alone included medication/prescription preparation, sterile compounding, problem-solving, and billing/inventory management. Likewise, certification in combination with previous work experience as a technician was viewed to have a more positive impact, with positive mean values (above scale median) for all items except for ethical decision-making. Many of the mean values were similar to but in some cases perhaps somewhat greater than those of certification combined with other education/training. Higher mean values were seen with regard to tech-check-tech, emerging responsibilities (e.g., administering immunizations and assistance with medication therapy management), and a few others; however, these were not compared statistically.

Table 3 provides mean ratings of items/factors contributing to making certification more impactful. Most items were evaluated quite highly on a three-point scale of importance, with all but one of them at or above the median scale value of “2”. The factor rated below “2” was “more difficult examination”. Items the respondents rated rather high on importance included better integration of the certification process with vocational training, more content on technical pharmacy knowledge/skills, more content on “soft skills”, and more support for certification from employing organizations.

Respondents’ beliefs about various facets of the value of certification are shown in Table 4. Respondents slightly disagreed with the notion that certification is equally beneficial across different practice settings. There was also slight disagreement with mean scores toward neutral for items suggesting certified technicians make fewer mistakes, are more innovative in customer service, are better prepared to deal with organizational change, and are more committed to their employer. There was agreement with the idea that technicians with experience are able to leverage certification, are more committed to their occupation/profession, help to promote a stronger organizational culture, and are better prepared to accept new roles, as well as that hiring decisions are made at least in part on whether the technician is certified. There was strongest agreement with the idea that technician certification should be a requirement for advanced status and/or roles.

Table 5 provides the mean ranking of various factors responding pharmacists actually use or would use in hiring pharmacy technicians. Ranked highest (lowest mean) was the technician’s demonstrated or anticipated job abilities. This was followed by their previous work history as a technician, their communication skills and moral integrity, whether or not they are certified, their ability to adapt to practice change, their emotional intelligence, and finally their acquisition of vocational school training.

## 4. Discussion

This study evaluated the opinions of pharmacists from four states regarding the value of and potential changes that might enhance the impact of pharmacy technician certification. In doing so, it updated previous assessments of certification undertaken over a decade ago after many changes in the pharmacy landscape and after continued calls for standardization in technician education and training. It also undertook this evaluation under the auspices of an organizational behavior framework, thus focusing more on general abilities that transcend practice setting while considering potential organization and further practice change.

Although approaching the topic from a different angle, namely with an organizational behavior framework, the study corroborates other research on pharmacist workforce, such as how technicians see themselves in regard to their own preparedness [29]. The results also align with a recent qualitative study of both pharmacists and technicians identifying a competency/preparedness for practice framework that identified six domains, including: communication in patient care, collaboration with other personnel, knowledge in pharmaceuticals, organization of care (including staffing and workflow issues), emerging leadership responsibilities, and personal development [36].

Pharmacists responding to the survey saw certification alone as having only a very modest impact on technician competence in various job responsibilities and behaviors. This is not surprising, given that certification involves a self-study process that does not include a didactic or experiential component and was basically designed to impart certain foundational knowledge concepts for test-takers [24]. As such, it also is not surprising that the competencies accorded the highest impact by certification were basic pharmacology and math computational skills. However, respondents saw certification as having a greater impact when it is combined with other types of education/training and with previous work experience. In comparison to certification alone, for certification in combination with other educational activities and/or work experience, the impact was evaluated much higher on competencies such as medication preparation, compounding, billing/administrative functions, problem-solving, and leadership. As such, the responding pharmacists likely recognize the importance of longitudinal and multiple types of exposure to more complex and cognitive functions administered or instructed in a variety of ways [37]. Likewise, there have been calls in Doctor of Pharmacy (PharmD) education to include various types of learning experiences for skills such as problem-solving, leadership, and managing change [38].

These results suggest that respondents see certification as an important component of a larger effort to promote technician competence and professionalization [39]. In fact, the action deemed most important by respondents in certification having an even greater impact was integration of the certification process with vocational training. This was echoed in the aforementioned stakeholder consensus conference and other calls for standardizing technician preparedness and entry into the field [18]. In the current study, pharmacists viewed technician vocational training as least impactful in potential hiring decisions, and this reflects concern pharmacists have expressed about the variation of quality in those vocational programs [40]. However, the pharmacists in the current study see certification’s alignment with vocational programs as a potential way to further standardize and raise the quality of technician preparedness for practice. These phenomena warrant additional study.

Respondents reported nearly the same level of importance on content related to technical pharmacy skills and to the inclusion of “soft skills” such as leadership and ethical decision-making. Indeed, the evaluation of the impact of certification alone on items such as communication and leadership were relatively low. Previous research has found community pharmacists of the mindset that technicians are the “face” of the pharmacy [41] and have also expressed concern about technicians dealing with controlled substances (i.e., issues around ethical decision making) [42]. Thus, while it is unlikely that anyone expects a self-study examination process to fully prepare technicians for specific job competencies and soft skills, respondents did think it was an important component to add and perhaps integrate with other types of education and training. This is further amplified by a high importance rating given to the need for more support from employers, which likely also alludes to the need to increase technician salaries as well as to better leverage certification in their own in-house training. Previous qualitative research indicates pharmacist support for including more so-called soft skills [43]. Viewed as less than somewhat important in this study was the need to simply make the current certification examinations more difficult.

Respondents reiterated their perceptions of the importance of communication skills and professional judgment when these skills were rated rather highly in hiring decisions, only after previous work experience and expected overall abilities on the job. Certification was deemed to be relatively important, and alone, was deemed more important than vocational training as well as the technician’s ability to adapt to practice change. Perhaps respondents believe that technicians’ ability to adapt to change and their emotional intelligence will, or can be groomed by the employing organization. The results here would appear to corroborate recent explorations into the desired characteristics of pharmacists, as well. Alston et al. [31] found characteristics such as communication skills and moral integrity to be the most highly sought after among recent pharmacy graduates for new positions, and Wheeler et al. saw these types of competencies, as well as proven ability and experience, to be among the most frequent requirements listed in pharmacist job positions posted nationwide [44].

A composite view of certification was undertaken through the use of general items about various facets of the process. Respondents agreed that technicians with experience are able to leverage certification effectively, thus adding to perceptions of its importance but need for integration with experience, which would appear to be incumbent both on employers and certification boards. The responding pharmacists also agreed that certification should be required for designation into advanced status, as is becoming more common with career laddering options [45]. Respondents also agreed that technicians who are certified are more committed to their profession but slightly disagreed or were neutral in regard to employer commitment. Perhaps pharmacists are of the belief that having gained more marketability through certification might result in technicians being open to opportunities with other employers. The respondents also slightly disagreed or were neutral with the notion that certified technicians are more adept in customer service and commit fewer errors. Again, certification is not necessarily meant to improve skills that would be associated with reduced errors, but perhaps better integration and more emphasis on public health/safety and customer relations would be beneficial components to the certification process.

Taken together, the results suggest that pharmacists placed a good bit of value on technician certification but are aware that certification alone does not prepare technicians for greater competence in all facets of work and that the role it plays currently is in providing much needed background knowledge in certain areas. The results also suggest that respondents see certification as a needed component of technician education and particularly required for advancement into higher status/roles in the organization and helping to imbue greater commitment and professionalism through the self-study process while acknowledging that certification needs to be better synchronized and leveraged with other experiences and education. As such, these results call for action by employers, certification organizations, educators, and other leaders in the profession concurrently, thus echoing sentiments expressed for quite some time [46,47]. Even colleges/schools of pharmacy in the U.S. can consider interprofessional education options that include exposure of technicians to Doctor of Pharmacy curricula while providing Doctor of Pharmacy students an opportunity to gain greater appreciation for technician roles, which can ultimately assist as well in their supervision of technicians and thus technicians’ competence [48]. Additionally, technician certification vendors (and educational organizations) might want to gear future education and assessment based more on the setting than on the particular task.

### Study Limitations

Several study limitations are worth noting. The survey was administered to pharmacists in only four U.S. states. The response rate achieved was positive in light of evidence suggesting that surveys, particularly those executed through email, are otherwise quite low [49]. However, the response rate was low enough to preclude generalization of attitudes even to pharmacists in the four states comprising the sampling frame. The research involved the use of question items informed from previous research but without any sort of gold standard or an attempt to create an overall composite measure or index; thus, there was no basis by which to discern the construct or content validity of the measures used.

## 5. Conclusions

This study examined pharmacists’ perceptions of the value of pharmacy technician certification within an organizational behavior framework. Pharmacists viewed certification as important for professionalization of technicians, suggested the need to include more soft skills training as part of the process, and implored leaders to better integrate certification with vocational training. Future research might attempt to replicate the current study with a more geographically diverse population and evaluate more specific strategies to integrate certification with other forms of training.

Even with many changes having taken place since initial evaluations of the value of certification, the results corroborate the importance of certification not only for specific skills but overall professionalism and commitment, yet recognizes the need for action by various stakeholders and leaders in pharmacy to better integrate certification with other educational components and past work experience to ensure a support workforce that has the competence needed to assist with the execution of effective patient care in a continuously changing health care environment.

## Figures and Tables

**Table 1 pharmacy-07-00147-t001:** Demographic and work-setting characteristics of responding pharmacists (*n* = 326).

Characteristic	Number (%) *
U.S. State residing/practicing	
California	112 (34.5%)
Florida	83 (25.4%)
Ohio	63 (19.3%)
Tennessee	68 (20.9%)
Sex	
Female	170 (52.2%)
Male	156 (47.8%)
Race/ethnicity	
White/Caucasian	222 (68.1%)
Black/African-American	18 (5.6%)
Hispanic/Latino	24 (7.5%)
Asian/Pacific Islander	15 (4.6%)
Middle Eastern (e.g., Arabic, Persian, Palestinian)	5 (1.4%)
Prefer not to answer	42 (12.9%)
Hours worked per week	
Up to 20	42 (12.9%)
20–39	43 (13.2%)
Greater than 39	241 (73.9%)
Primary work setting	
Community independent	86 (26.3%)
Community chain	91 (27.9%)
Community health center (e.g., Federally Qualified Health Center)	2 (0.6%)
Hospital inpatient	49 (15.0%)
Hospital ambulatory care	12 (3.7%)
Hospital critical access	7 (2.1%)
Hospital other	6 (1.8%)
Compounding/other specialty	22 (6.7%)
Government state or local	9 (2.8%)
Government federal	7 (2.1%)
Government military	6 (1.8%)
Mail service	7 (2.1%)
Managed health care	12 (3.6%)
Pharmaceutical industry	4 (1.2%)
Other	6 (1.8%)
Job title	
Staff pharmacist	100 (30.7%)
Clinical pharmacist	71 (21.6%)
Pharmacy manager or supervisor	79 (24.2%)
District manager	2 (0.7%)
Chief pharmacist/Pharmacy Director/Assistant Director or Chief	28 (8.6%)
Pharmacy Owner	48 (14.7%)

* Percentages for each category may not summarily equal 100.0% due to rounding.

**Table 2 pharmacy-07-00147-t002:** Perceptions of the impact of certification on technician competence alone, in combination with other types of education/training, and in combination with previous work experience.

Skill/Behavior/Knowledge Item	Alone *	with Education *	with Work *
Medication order/prescription entry	2.38±1.24	3.54±1.42	3.65±1.35
Medication order/prescription preparation	2.15±1.07	3.46±1.49	3.62±1.18
Patient/customer service	1.92±1.17	3.15 ± 1.33	3.27 ± 1.37
Non-sterile compounding	2.35 ± 1.24	3.58 ± 1.42	3.65 ± 1.37
Sterile compounding	2.19 ± 1.00	3.54 ± 1.34	3.65 ± 1.36
Inventory management	1.92 ± 0.91	3.38 ± 1.42	3.46 ± 1.34
Billing and other administrative functions	1.73 ± 0.92	3.45 ± 1.30	3.53 ± 1.42
Interpersonal communication	1.65 ± 0.98	3.00 ± 1.36	3.02 ± 1.19
Time management/organization skills	1.73 ± 1.12	2.92 ± 1.47	3.04 ± 1.32
Ethical decision making	1.92 ± 1.22	3.04 ± 1.45	2.88 ± 1.40
Supervision of other technicians	2.19 ± 1.08	3.31 ± 1.40	3.42 ± 1.35
Tech-check-tech	2.31 ± 1.12	3.27 ± 1.31	3.46 ± 1.28
Quality assurance program activities	2.19 ± 1.15	3.38 ± 1.28	3.50 ± 1.24
Professionalism	2.12 ± 1.12	3.23 ± 1.37	3.27 ± 1.33
Problem-solving/innovativeness	1.88 ± 1.01	3.23 ± 1.29	3.37 ± 1.28
Accepting responsibility	1.88 ± 1.21	3.12 ± 1.28	3.15 ± 1.30
Leadership	1.77 ± 1.29	3.24 ± 1.37	3.38 ± 1.34
Emerging/new responsibilities (e.g., immunizations, assistance with MTM)	2.19 ± 1.28	3.48 ± 1.36	3.52 ± 1.33
Basic pharmacology/knowledge of drug names/OTCs	2.69 ± 1.34	3.58 ± 1.36	3.65 ± 1.31
Math computational skills	2.65 ± 1.20	3.65 ± 1.34	3.58 ± 1.28
Managing organizational change	2.08 ± 1.27	3.08 ± 1.28	3.12 ± 1.19

* Mean ± Standard deviation on a scale ranging from 1 = Not at all to 5 = Very much.

**Table 3 pharmacy-07-00147-t003:** Items/factors contributing to making certification more impactful.

Item/Factor	Mean ± S.D. *
More difficult examination	1.83 ± 0.69
More stringent criteria to sit, or be able to take the examination	2.00 ± 0.76
More specialty certifications (e.g., specifically in compounding, Inventory management)	2.00 ± 0.87
More process/logistical support from certification programs (PTCB, NHA)	2.05 ± 0.91
More support from your employing organization for certification	2.35 ± 0.74
More content on “soft skills” such as communication, leadership, and ethical decision-making	2.39 ± 0.65
More content on technical pharmacy knowledge and skills	2.45 ± 0.62
Better integrating the certification process with vocational training	2.50 ± 0.64

* Mean ± Standard Deviation on a three-point scale ranging from 1 = Not important to 3 = Very important.

**Table 4 pharmacy-07-00147-t004:** Respondents’ beliefs about various facets of the value of certification.

Item/Facet	Mean ± S.D. *
Certification imparts the same level of benefit to technicians regardless of setting	2.83 ± 1.51
Technicians with experience are able to leverage certification effectively	4.57 ± 1.24
Certification assists technicians who are new to this field of work	3.55 ± 1.57
I make (or would make) technician hiring decision at least in part upon whether or not they are certified	3.99 ± 1.75
Technician certification should be a requirement for advanced status/roles	4.84 ± 1.63
Technicians who are certified are better prepared to accept new roles and responsibilities	3.98 ± 1.84
Certified technicians make fewer mistakes/errors than non-certified ones	3.08 ± 1.56
Technicians who are certified are better prepared to deal with organizational change	3.25 ± 1.60
I feel more comfortable delegating to a technician who is certified	3.82 ± 1.80
Technicians who are certified help to promote a stronger organizational culture in my organization	3.77 ± 1.72
Technicians who are certified are more innovate in providing customer/client service	3.23 ± 1.60
Technicians who are certified are more committed to their employer	3.13 ± 1.68
Technicians who are certified are more committed to their occupation/profession	4.34 ± 1.75

* Mean ± standard deviation on a scale ranging from 1 = Strongly disagree to 6 = Strongly agree.

**Table 5 pharmacy-07-00147-t005:** Mean ranking of factors in respondents’ hiring or potential hiring decisions regarding pharmacy technicians.

Item/Factor	Mean ± S.D. *
Their demonstrated or anticipated job abilities	2.97 ± 1.78
Their previous work history as a technician	3.53 ± 1.37
Their communication skills	3.65 ± 1.64
Their professional/moral integrity	3.65 ± 1.79
Whether they are certified	4.54 ± 2.35
Their ability to adapt to practice change	5.61 ± 1.69
Their emotional intelligence	5.83 ± 2.14
Their acquisition of vocational school training	6.22 ± 1.97

* Mean ± standard deviation based on a ranking of each item/factor from 1 to 8.

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
