# Peer review of "The Value and Potential Integration of Pharmacy Technician National Certification into Processes That Help Assure a Competent Workforce"

_pharmacy, 2019, doi:10.3390/pharmacy7040147_

Round 1
Reviewer 1 Report
This paper describes the results of a survey of pharmacists collecting their perceptions on the value of pharmacy technician certification relative to other training and education as well as other job skills. Overall, the paper reads clearly but deserves a final check for grammar and typographical errors, particularly spacing. The results will be of interest to the journal's readers and deserve publication once minor revisions are made.
Introduction: No changes
Materials and Methods: The sourcing of questions, particularly for questions described in lines 139-160, is vague. Clarity would be appreciated. In particular, on line 155 the authors refer to a previous item related to hiring decisions, but there is no citation for this previous item.
Results: Some rough comparison to pharmacist demographics would be helpful to get a sense of nonresponse bias. Not a critical element, but if demographic data can be sourced, for example from the Bureau of Labor Statistics, this would be helpful in building confidence in the results.
Line 223: "media" is used when I believe you meant "median"
Discussion: Line 295-297: The conflict between needing integration of certification into vocational training and vocational training being rated the lowest priority item on Table 5 deserves additional exploration. I don't see this in the results section.
Tables and Figures: Table 3: It would be helpful to reorder this by mean to aid the reader's interpretation of the responses.
Author Response
This paper describes the results of a survey of pharmacists collecting their perceptions on the value of pharmacy technician certification relative to other training and education as well as other job skills. Overall, the paper reads clearly but deserves a final check for grammar and typographical errors, particularly spacing. The results will be of interest to the journal's readers and deserve publication once minor revisions are made.
Thank you very much for your encouraging feedback! We apologize for typos and such. A new Word version on a new computer for some reason was 'sticking words together'. We've tried to the extent possible to rectify that. Though this is not a proper 'excuse' for these and other areas. Again, we've gone through the paper a few more times and attempted to clean everything up.
Introduction: No changes
Materials and Methods: The sourcing of questions, particularly for questions described in lines 139-160, is vague. Clarity would be appreciated. In particular, on line 155 the authors refer to a previous item related to hiring decisions, but there is no citation for this previous item.
We have cited sources for our methods questions. And as stated in the limitations there is no gold standard here for measuring any of these things, and these things largely have not been strictly measured. Thus, we took 'inspiration' from mention of these concepts and previous papers that indirectly or crudely measured them. These involved some references we already had, but also a couple new ones now in the revised manuscript, including preferred job characteristics of pharmacists (not technicians) by Alston et al.
Results: Some rough comparison to pharmacist demographics would be helpful to get a sense of nonresponse bias. Not a critical element, but if demographic data can be sourced, for example from the Bureau of Labor Statistics, this would be helpful in building confidence in the results.
The BLS has data on pharmacist work setting, and it is indeed very broad categorizations. But we did include this and compared it favorably to the work setting breakdown of our sample. We also found responses by gender to a national survey conducted in 2014 by Schommer et al and have included a comparison to those, as well, in a new citation in the revised draft.
Line 223: "media" is used when I believe you meant "median"
Thank you. Fixed.
Discussion: Line 295-297: The conflict between needing integration of certification into vocational training and vocational training being rated the lowest priority item on Table 5 deserves additional exploration. I don't see this in the results section.
We now expound upon this a bit further, clarifying and citing that vocational education has been deemed programmatic and inconsistent. As such, integration with certification or aligning vocational education with certification might be one mechanism of assuring standardization and preparation of technicians entering the field and have rearranged a couple citation to underscore this, yet also state that this requires further study.
Tables and Figures: Table 3: It would be helpful to reorder this by mean to aid the reader's interpretation of the responses.
Done.
Reviewer 2 Report
I really enjoyed reviewing your manuscript. My comment for minor revision is following;
Please correct the reference format to be fitted with Pharmacy Journal.
Otherwise, well-written.
1. The authors used self-administered survey for this study. However, I did not see any validated reference for using self-administered survey. What previous research has validated this survey instruments for measuring perception of pharmacy workforce? 2. In discussion, please discuss previous related literatures in other surveys related with pharmacy workforce. 3. Conclusions need to summarize the study results, pointing out the added values to the previous literature and future research area.
Author Response
Thank you for your comments and suggestions, as well as your support of the paper. Below are changes made/responses to your recommendations.
The authors used self-administered survey for this study. However, I did not see any validated reference for using self-administered survey. What previous research has validated this survey instruments for measuring perception of pharmacy workforce? Thank you for pointing this out. And your comments were very similar to those of the other reviewer. There is not previously validated research in this area. There are comments and calls to action, including from the recent stakeholder conference referenced in the paper. There are also other studies that get at similar constructs but not measured in this same way, ie, under an organizational behavior context. Still, there were several studies that inspired creation of the current items. All but one were already cited and elsewhere, and we've 're-cited' them, here, also with the inclusion of a new citation (Alston et al) that measured similar constructs in pharmacists. That study is recent but has been online for over 1.5 years and did serve as an inspiration for some items, specifically attributing value to various traits/behaviors/experiences in hiring decisions In discussion, please discuss previous related literatures in other surveys related with pharmacy workforce. 3. Conclusions need to summarize the study results, pointing out the added values to the previous literature and future research area.Thank you. We have added a couple recent studies that examined technician workforce, including a very new one that used qualitative measures combining the responses of both pharmacists and technicians to proffer competence/preparedness frameworks for future practice. We have made additions to the Conclusion section to highlight key findings of integrating certification with other forms of training and future research needed to evaluate different modalities in which this might take place, along with the need to replicate the current study with a more geographically diverse population.